# Hierarchical organic microspheres from diverse molecular building blocks

Yintao Li[1], Longlong Fan [2], Xinyan Xu[3], Yang Sun[1], Wei Wang[1], Bin Li [3], Samuel S. Veroneau [4] & Pengfei Ji [1] ✉

Microspherical structures find broad application in chemistry and materials science, including in separations and purifications, energy storage and conversion, organic and biocatalysis, and as artificial and bioactive scaffolds. Despite this utility, the systematic diversification of their morphology and function remains hindered by the limited range of their molecular building blocks. Drawing upon the design principles of reticular synthesis, where diverse organic molecules generate varied porous frameworks, we show herein how analogous microspherical structures can be generated under mild conditions. The assembly of simple organic molecules into microspherical structures with advanced morphologies represents a grand challenge. Beginning with a partially condensed Schiff base which self-assembles into a hierarchical organic microsphere, we systematically synthesized sixteen microspheres from diverse molecular building blocks. We subsequently explicate the mechanism of hierarchical assembly through which these hierarchical organic microspheres are produced, isolating the initial monomer, intermediate substructures, and eventual microspheres. Furthermore, the open cavities present on the surfaces of these constructs provided distinctive adsorptive properties, which we harnessed for the immobilization of enzymes and bacteriophages. Holistically, these hierarchical organic microspheres provide an approach for designing multi-functional superstructures with advanced morphologies derived from simple organic molecules, revealing an extended length scale for reticular synthesis.

Microspheres emerge in biological and materials sciences as versatile constructs with respect to both structure and function. The microspherical structures of pollen give rise to its advantageous adhesive properties and bioactivities[1]. Microspherical $CaCO_3$ particles evolved in specialized cells of alpine plants enhance light harvesting[2]. Synthetically, porous microspheres have been applied to catalysis[3–5], energy storage[6–8], and biomedical applications[9–11], owing to their high surface-area-to-volume ratios, dispersibility, and adsorption properties. Polystyrene microparticles, for example, are common platforms for immunoprecipitation and related techniques. Such synthetic microspheres are commonly prepared from organic polymers, including polystyrene[4] and poly-ferrocenyldimethylsilane[12,13], or inorganic compounds, such as porous carbon[6,14], silica[15,16], metals[17,18], and metal complex[5,7,19]. Extending the structures and functions accessed by these architectures, however, will require expanding the diversity of these compositional building blocks.

In the field of porous materials, organic precursors give rise to diverse types of covalent organic frameworks (COFs) through a

[1]Department of Chemistry, Zhejiang University, Hangzhou 310058, China. [2]Institute of High Energy Physics, the Chinese Academy of Sciences, Beijing 100049, China. [3]College of Agriculture and Biotechnology, Zhejiang University, Hangzhou 310058, China. [4]Department of Chemistry and Chemical Biology, Harvard University, Cambridge, MA 02138, USA. ✉e-mail: jipengfei@zju.edu.cn

combination of covalent bonds and intermolecular interactions (Fig. 1a)[20–22]. Exploiting the interactions of organic building blocks to tailor the physical and structural characteristics of framework materials was also demonstrated in the design and synthesis of hydrogen-bonded frameworks (HOFs, Fig. 1b) among other materials. HOFs similarly assemble from molecular building blocks, though through non-covalent interactions, further diversifying the molecular interactions begetting porous materials[23,24]. These materials include a range of molecular building blocks and interactions that accommodate a commensurately wide range of physical (e.g., adsorption) and structural (e.g., pore size) properties[25–30]. By leveraging similar covalent and hydrogen-bonding interactions, we demonstrate how modular synthesis may be extended into constructing diverse microspherical superstructures from diverse molecular building blocks.

Here we report a series of hierarchical organic microspheres (HOMs) assembled from V-shaped organic molecules through hydrogen-bonding interactions and π-π stacking. To the best of our knowledge, such non-polymeric organic microspheres are rarely reported in the literature, though are readily synthesized herein through straightforward solvent-driven processes. Furthermore, these assemblies generate large open cavities that may be employed in adsorptive and catalytic processes. From the sixteen different structures that were assembled from varied molecular building blocks, we propose that the hierarchical self-assembly of primary molecular building blocks proceeds through secondary structures of either

threads or sheets, depending on the relative strength and directionality of π-π stacking and hydrogen bonding. These secondary structures are then proposed to interact and form either echinate microspheres, through the winding of one-dimensional threads, or striated microspheres, through the interpenetration of two-dimensional sheets (Fig. 1c). These iterative steps are affected by the type of organic building blocks and the synthetic conditions, allowing for modulation of particle sizes, cavity sizes, and surface properties. Importantly, the hydrogen bonding responsible for these (sub)structures improves the solution processibility of HOMs[24,26,31] whereby myriad small molecules could generate a range of microspheres with increasingly diverse morphology and function[31,32]. With respect to function, these sixteen HOMs displayed very different properties for potential applications in enzyme immobilization and phage protection. Holistically, this study introduces the principles of reticular synthesis to the rational design and synthesis of microspheres, motivating future efforts into how microspheres and other microscale architectures may be generated from organic building blocks.

## Results
### Synthesis and structure of HOM-1
The reaction of 2,4,6-trihydroxy benzene-1,3,5-tricarbaldehyde (TP, **1**) with 4-aminobenzoic acid (ABA, **2**) in dimethylacetamide (DMAc) generates a triple condensation that precipitates (TACT, **3**) as a partially crystalline powder with no microscale structure (Supplementary

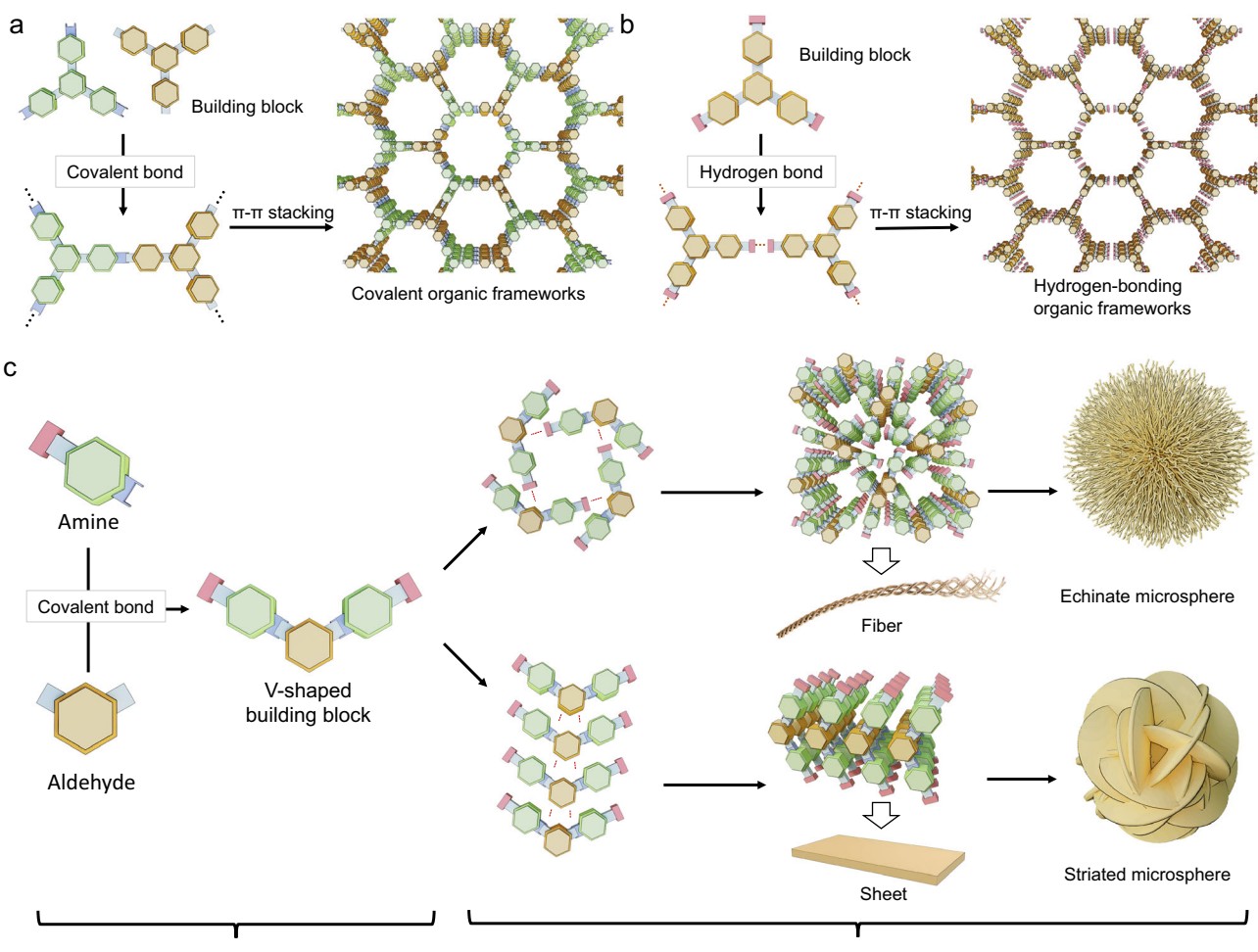

**Fig. 1 | Structural feature of hierarchical organic microspheres in comparison with classical porous organic materials.** Scheme of the assembly mechanism of covalent organic frameworks (**a**) and hydrogen-bonded organic frameworks (**b**).

**c** Scheme on the design of V-shaped molecular building blocks and the proposed self-assembly mechanism to form hierarchical organic microspheres.

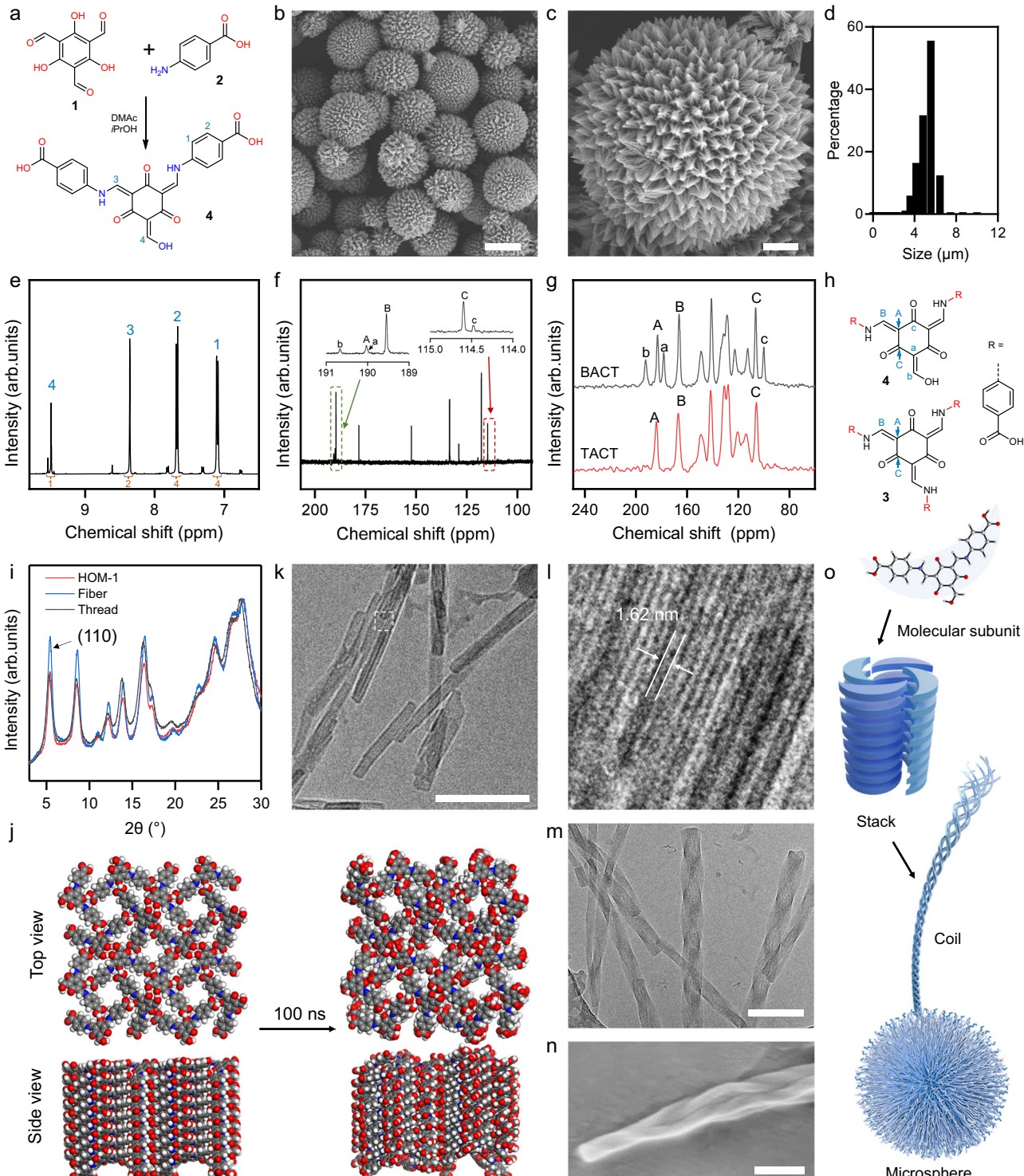

**Fig. 2 | Optimization and synthesis of HOM-1. a** Scheme of the double condensation reaction of **1** and **2**. **b, c** SEM of HOM-1 (the left scale bar is 4 μm and the right scale bar is 1 μm). **d** Size distribution of HOM-1 measured from DLS. **e** ¹H NMR spectrum of **4**. **f** Liquid-state ¹³C NMR spectrum of **4**. **g** Solid-state ¹³C NMR spectrum of HOM-1 (BACT) and **3** (TACT). **h** Structural differences between **3** and **4**. Carbon signals from the TP motif are assigned. **i** PXRD pattern for **4** assembled in different solvent combinations. **j** MD simulation for the ideal model of HOM-1 containing 128 pre-ordered **4** molecules in the DMSO-MeOH (2978 molecules for each) combination. Cryo-EM image of thread (**k**) and measurement of lattice spacing (**l**). The scale bar is 100 nm. **m** Cryo-EM image of entangled fibers. The scale bar is 100 nm. **n** SEM image of entangled fiber. The scale bar is 100 nm. **o** The four main stages during the hierarchically assembled process of **4** through HOM-1.

Fig. 1). This same reaction in DMAc and isopropanol (iPrOH) produces a double condensation product (BACT, **4**) that instead forms crystalline particles of echinate spherical morphologies (Fig. 2a–c). The size distribution of these spheres was analyzed with dynamic light scattering (DLS), giving an average particle size of 5.5 ± 0.9 μm (Fig. 2d).

This material is described as a hierarchical organic microsphere (HOM) and this prototypical construct is numbered HOM-1.

The chemical composition of HOM-1 was determined through multiple analytical techniques (Fig. 2e–h). When dissolved in a 14 mM aqueous solution of ammonia, the mass of the produced species is

consistent with that of **4** (found: 447.0837 Da [M-H]$^-$ and calculated: 447.0834 Da [M-H]$^-$). When dissolved in 23 mM NaOD in $D_2O$, $^1H$ and $^{13}C$ NMR spectra can be obtained; $^1H$ NMR spectrum show four peaks that integrate in a 1:2:4:4 ratio and $^{13}C$ NMR spectrum showed 11 peaks, consistent with that expected for **4** (Fig. 2e). Comparing solid-state and solution $^{13}C$ NMR spectra (Fig. 2f), there is an upfield shift in the solid state because subunits were protonated, though the same number of peaks is observed[33] corroborating HOM-1 to be composed of the double condensation product. The solid-state $^{13}C$ NMR of **3** was collected as a reference. The three signals were observed for the TP moiety, exhibiting three singlet peaks at chemical shifts of 183.0, 166.1, and 106.3 ppm (labeled as ABC in the brown line, Fig. 2g, h). In comparison, the solid-state $^{13}C$ NMR spectra of HOM-1 showed splitting of the three carbon signals into six, with the appearance of three additional peaks with relatively lower intensities (labelled as abc in the dark line, Fig. 2g). This phenomenon was caused by the incomplete condensation of one of the three aldehyde groups, which exhibits a different chemical environment from the condensed imine group, consistent with the proposed double-condensation structure in the HOM-1 material. The condensation was also confirmed by the disappearance of C = O stretching in the FT-IR spectrum of HOM-1 (Supplementary Fig. 2).

The underlying structure of HOM-1 was solved from powder X-ray diffraction (PXRD) spectra in combination with computational methods (Fig. 2i, Supplementary Figs. 3–6). Indexing of the measured PXRD pattern of HOM-1 suggested a tetragonal unit cell with dimensions of 23.05 Å × 23.05 Å × 3.94 Å in the P4 space group. The complete structure of HOM-1 was solved by refining PXRD patterns by the Rietveld method, affording simulated PXRD patterns that corresponded with those measured experimentally (Supplementary Method 5, Supplementary Fig. 7). This extracted structure implicated a hydrogen-bonding network generating from underlying V-shaped building blocks. In our proposed model, each molecule hydrogen bonds with at least five neighboring molecules mainly through the interactions between carboxyl and β-aminoenone groups (Supplementary Fig. 5d). As well, interlayer stacking governed by parallel-displaced π–π interactions (Supplementary Fig. 6) promotes the formation of higher-order structures (*vide infra*). To assess the validity of these proposed interactions[34] we performed molecular dynamics (MD) simulations to test the stability of the modeled structure in the relevant solvent system (Fig. 2j). After 100 ns of simulation, the predicted hydrogen bonds and interlayer π-π interactions remained unchanged, maintaining the threads observed experimentally. These models further indicated nanoscale channels, justifying experimentally measured $N_2$-sorption and $CO_2$-sorption isotherms (Supplementary Fig. 8).

Further SEM analysis of the surface of HOM-1 revealed the barbs of these microspheres were themselves composed of coiled bundles of one-dimensional threads (Supplementary Fig. 9). Isolated and coiled threads could furthermore be separated from these syntheses (Fig. 2k–n). The diffraction patterns for these threads matched that of the microspheres, suggesting these materials to be structurally related. Cryo-electron microscopy of these threads corroborated these findings, with lattice fringe spacings (1.62 nm) were identical to the interplanar spacing of (110) planes from PXRD patterns of microspheres (Fig. 2i, l). This hierarchical process of forming threads and eventually microspheres is examined comprehensively below (Fig. 2o) and broadly includes: (1) molecular subunits that stacked through hydrogen bonding to form individual threads (primary structure), (2) threads which coil through further hydrogen-bonding and π- π stacking (secondary structure), and (3) microspheres that form through further branching and interaction of these coiled threads (tertiary structure). This final process is interrogated specifically below, whereby the progression and interaction of these threads can be inhibited or enhanced.

## Morphological diversity derived from synthetic protocols

The relevance of hydrogen-bonding to the morphology of HOM-1 implies that modulating the polarity and protic nature of solvent during synthesis may affect assembly (Fig. 3a–g). The assembly of HOM-1 was thus evaluated in varied synthetic conditions through high-throughput screening with 96-well plates (Fig. 3a–d, Supplementary Figs. 10–13). DMAc is an aprotic polar solvent whereas iPrOH is a protic solvent resulting in microspherical HOM-1 (Fig. 3). This precipitation was gradual (-15 min.) and could be quantified through scattering at 600 nm (Fig. 3h, dark line). Replacing iPrOH with increasingly polar solvents, as measured by relative polarity ($p$)[35], significantly changed both the rate of precipitation and morphology of HOM-1. Replacing iPrOH ($p = 0.546$) with ethanol (EtOH; $p = 0.654$) shortened the precipitation time to 10 min. and produced particles with inhomogeneous morphology (Fig. 3b1, h, red line). With methanol (MeOH; $p = 0.762$), this time was further shortened to 5 min. and produced predominately intertwined fibers with limited branching structures (Figs. 3c1, 3h, blue line). With water ($p = 1$) this trend continued, with immediate precipitation (<1 min.) of exclusively isolated threads 15 nm in diameter (Fig. 3d1, h, green line). (*N.B.*, the optical density of this water-DMAc system is lower than that of the other three tested systems as the precipitate forms a semi-transparent hydrogel.)

When DMAc ($p = 0.377$) is replaced with less polar aprotic solvents—ranging from N-methyl-2-pyrrolidone (NMP; $p = 0.355$) to toluene ($p = 0.099$)—microspherical HOM-1 could all be achieved when iPrOH was maintained as the protic solvent (Fig. 3a). As well, isolated fibers of HOM-1 were achieved regardless of aprotic solvent polarity when MeOH and $H_2O$ were included. Slightly less polar solvents, such as dichloromethane ($p = 0.309$) and NMP, produced microspherical HOM-1 with EtOH. Dramatically less polar aprotic solvents such as tetrahydrofuran (THF, $p = 0.207$) afforded a branched-rod structure with a diameter of each rod of about 1 μm when combined with EtOH (Fig. 3b4). Finally, hollowed nanorods with larger diameters of 3 μm and pore sizes of 1 μm could also be observed in THF, suggesting more advanced constructs might be accessible (Supplementary Fig. 15).

The morphology of HOM-1 was further affected by the ratio of solvents (Fig. 3e). When the volume fraction of iPrOH is above 80% in DMAc, the rate of precipitation increases significantly (Fig. 3i) and results exclusively in the thread morphology of HOM-1 (Fig. 3e). When this fraction is below 20%, HOM-1 assembly does not proceed, likely due to the increased relative polarity and thus solubility in the solvent system (Fig. 3i green line). We further investigated how the concentration of substrates affected HOM-1 assembly (Fig. 3f). Two-fold dilution (10 mM) or concentration of **1** and **2** (40 mM) had minimal impact on the morphology of HOM-1. Further dilution (<5 mM) prevented microspherical HOM-1 assembly, with no increase in scattering being observed (Fig. 3j). Further concentration (>40 mM) accelerated the precipitation process resulting in inhomogeneous particles within 5 min. Finally, increasing the stoichiometric ratio between **1** and **2** significantly decreased the rate of precipitation (Fig. 3g, k, blue line); decreasing this ratio slightly increased the rate of precipitation while avoiding the formation of a triple condensation product (Fig. 3g, k, red line).

Cumulatively, these results corroborate the apical role of hydrogen-bonding predicted in the hierarchical generation of HOM-1 and the diversity of its morphologies. In highly polar or protic solvent systems, such as with $H_2O$ (Fig. 3d) or with a high concentration of iPrOH (Fig. 3e), HOM-1 assembly stalls at threads; the hydrogen-bonding capacities of these solvent systems stabilize these threads, disfavoring further hydrogen-bonding between them. The formation of threads is rapid (Fig. 3h, green line, Fig. 3i) suggesting this stage precedes the higher-order morphologies which develop over longer timescales. As the polarity and protic nature of these solvent systems decrease, hydrogen-bonding between these threads becomes more

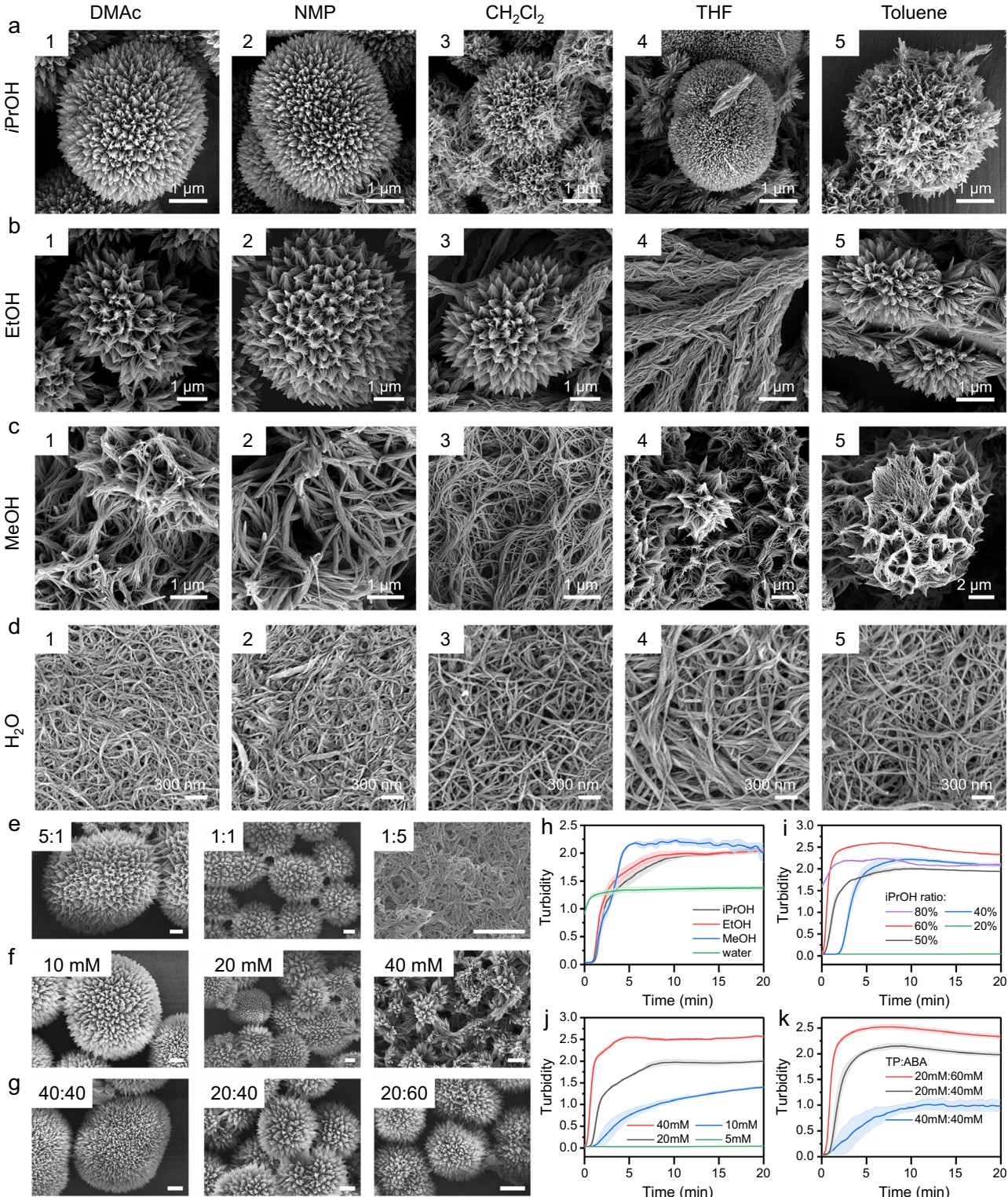

**Fig. 3 | The morphology of HOM-1 influenced by growth condition.**
**a**–**d** Morphologies of assembly of **4** synthesized in different solvent combinations. Effect of solvent ratio (DMAc:*i*PrOH, **e**), concentration (**f**), and reactant ratio (**1:2**, mM:mM, **g**) on the morphologies of assembly. The scale bars are both 1 μm. Effect of solvent type (**h**), solvent ration; (**i**), concentration (**j**), reactant ratio (**k**) on the precipitation kinetics. Error area represents standard deviation, $n = 3$.

favorable as do π-π interactions. On longer timescales, higher-order morphologies are thus able to develop, including the microspherical assembly architectures initially observed. The rate of this precipitation (Fig. 3h) and the specific morphology observed depends on the character of both solvents, with. for example, EtOH generating microspherical or tubular structures with DMAc and THF, respectively. This central role of solvent is evidenced as well by the minimal effects changing the concentrations (Fig. 3f) and ratios of precursors (Fig. 3g) have on HOM-1 morphology. Temperature also has significant influence on both the assembly dynamics and morphology of the HOM-1, with 25 °C being the optimal temperature for synthesizing the material (Supplementary Fig. 16).

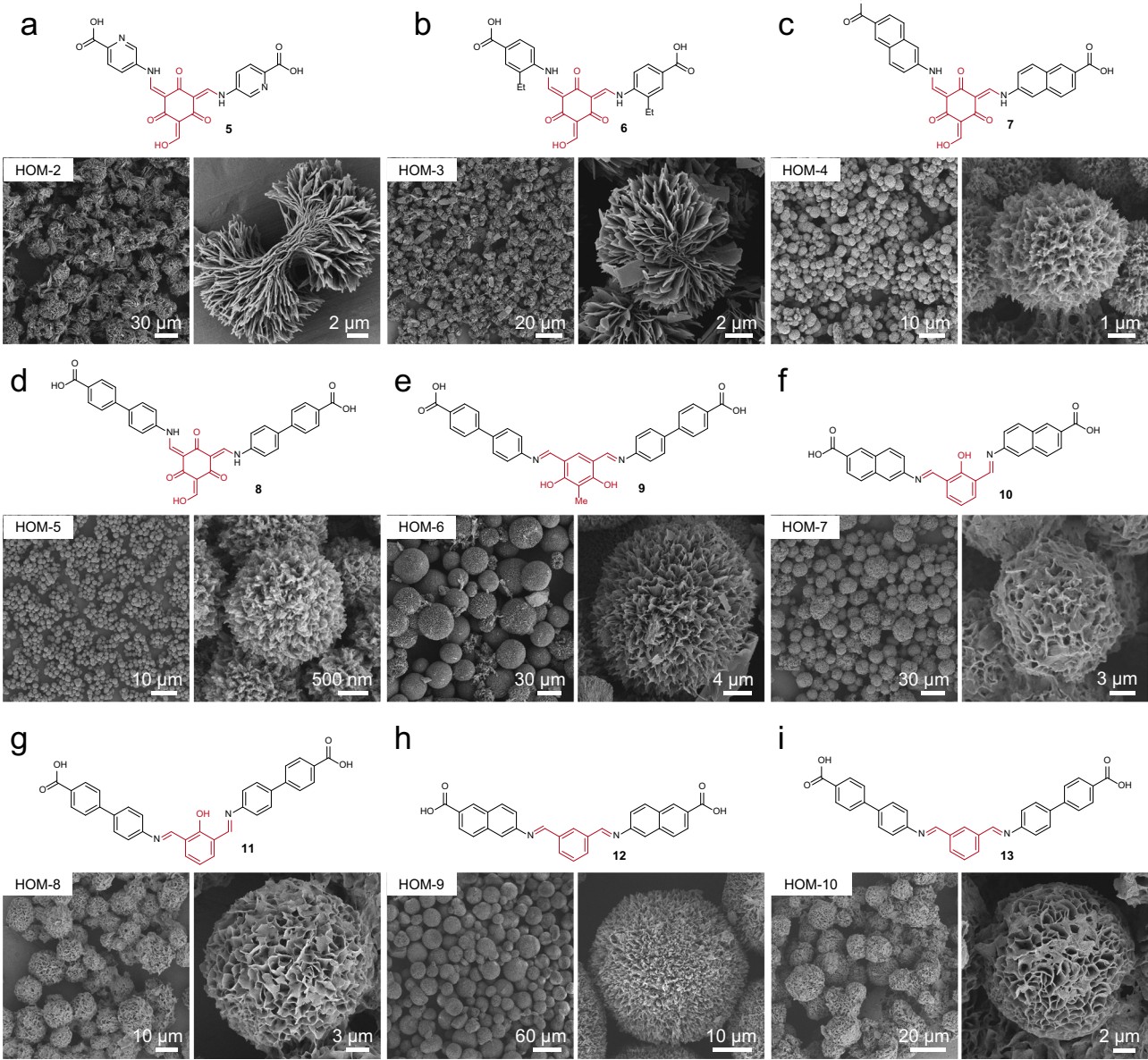

**Fig. 4 | Morphological diversity derived from diverse aryl dialdehydes. a–i** SEM images for the various HOMs based on varied aryl dialdehydes and amines. The synthetic condition for the HOMs is listed in Supplementary Table 1.

## Morphological diversity from assembled subunits

Microspherical HOM-1 assembly appears to be directed by hydrogen-bonding interactions between organic precursors that are favored or disfavored through solvent effects. Thus, V-shaped molecular building blocks with similar hydrogen-bonding propensities might generate diverse microspheres from these similar conditions. Retaining the central geometry of **4**, we obtain a series of HOMs, designated HOM-2 through HOM-16 through systematic screening of reaction solvents and temperatures (Figs. 4a–i, 5a–f and Supplementary Fig. 17). The compositions of these materials have been confirmed to be double condensation products through NMR and mass spectrometry (Supplementary Fig. 18–32). When the phenyl group of **4** is extended to naphthyl or biphenyl, microspherical structures of intertwined fibers could be synthesized from similar conditions as revealed through SEM (Fig. 4c, d). When an ethyl group is appended to the 4-carboxylphenyl substituent, the fibers convert to sheets while maintaining an overall microspherical structure through intercalation (Fig. 4b). To validate the necessity of the carboxyl group, we tested 4-hydroxylaniline and

4-nitroaniline for condensation with **1**. Neither of the two reactions afforded any precipitate, presumably for lacking hydrogen-bonding interactions from carboxyl groups.

Microspherical morphologies could furthermore be achieved when exchanging the core ligand with meta-substituted dialdehydes, whereby the triple condensation product is made entirely inaccessible. These ligands more strongly influenced the morphology of the materials, accessing HOMs with two distinct morphologies: (1) echinate microspheres assembled from one-dimensional threads (Figs. 4e, h, and 5f) and (2) striated microspheres assembled from two-dimensional sheets (Figs. 4f, g, and 5a–e). Among the sixteen HOMs obtained, HOM-9, produced from m-phenylenedialdehyde and 6-amino-2-naphthoic acid, exhibits the largest particle size with an average diameter of approximately 40 µm (Fig. 4h). In addition to these expected microspherical assemblies, several HOMs exhibit very different morphologies. HOM-11, produced from pyridine-2,6-dicarbaldehyde and 6-amino-2-naphthoic acid, exhibited a racemic striated structure, suggesting homochiral analogues might be

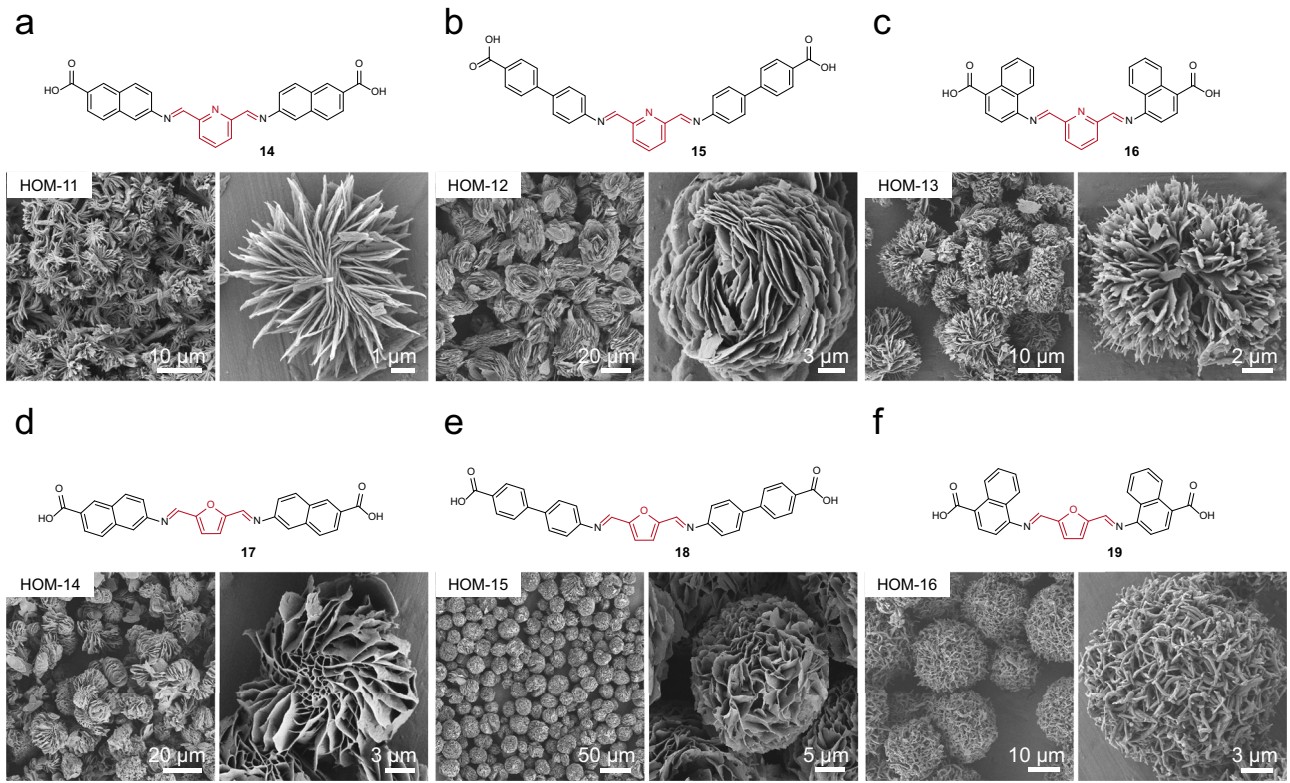

**Fig. 5 | Morphological diversity derived from diverse heteroaryl dialdehydes. a–f** SEM images for the various HOMs based on varied heteroaryl dialdehydes and amines. The synthetic condition for the HOMs is listed in Supplementary Table 1.

realized. HOM-14, produced from furan-2,5-dicarbaldehyde, yielded a striated morphology of stacked sheets that formed a central cavity (Fig. 5d). Finally, HOM-7, HOM-8, HOM-10, and HOM-15 all possessed network-like grooves distributed on their surfaces, highlighting the diversity of HOMs may formed from simple structural unit and solvent-driven processes. It is noteworthy that for the molecular subunit of HOM-15 (**18**), we managed to solve a single crystal structure, for which the simulated PXRD pattern differs from the experimental pattern of HOM-15, indicating that the stacking form of **18** in HOM-15 is different from that in the single crystal (Supplementary Fig. 33). Nevertheless, the structure reveals clear hydrogen-bonding and π-π stacking between the molecular units, indicating the potential role of such intermolecular interactions in the assembly of microsphere.

The stability of all the HOMs were tested under various environmental conditions. All the 16 HOMs showed high stability after long-term storage for one year (Supplementary Fig. 34), and thermal treatment, with decomposition temperatures of above 240 °C (Supplementary Fig. 35). All HOMs are stable in acid conditions (pH 1, HCl), while HOM-10, HOM-12, and HOM-15 are further stable in basic conditions (pH 12, NaOH) perhaps owing to their extended aromatic structure and attendant increase in interlayer π-π interactions (Supplementary Figs. 36 and 37).

## Modeling the geometric parameters of HOMs

The diversity of HOMs described herein requires statistical analyses to contrast the varied growth conditions and molecular subunits. Unlike traditional framework-based porous materials such as MOFs, HOMs exhibit type-II isotherms and thus the presence of both nanoscale and microscale cavities, precluding traditional Brunauer–Emmett–Teller (BET) calculations (Supplementary Fig. 38). We therefore required new parameters to describe the geometric features of HOMs. The tip-to-tip distances of surface features (hereafter point spacing) were thus

analyzed, which would determine the mass-transport properties of nanoscale pores (Supplementary Fig. 39).

First, we directly measured the distances between neighboring points, defined as: three points which form a triangle such that no fourth point lies within its circumcircle. Second, we partitioned a given set of points into non-overlapping triangles employing the Delaunay triangulation method[36] (Supplementary Fig. 40). Third, and finally, the edge lengths of the resulting triangles were calculated and considered as the point spacing. In this way, we analyzed a single HOM-1 particle and obtained an average point spacing of $0.493 \pm 0.167 \, \mu m$ (Fig. 6a). Subsequently, we analyzed a collection of more than 40 HOM-1 particles and calculated an average point spacing of $0.476 \pm 0.154 \, \mu m$ (Fig. 6b). To corroborate this approach, we separately calculated the point spacing using these same SEM images and computed radial distribution functions (RDF) of the point sets (Supplementary Fig. 41)[37,38]. The peak of this function was found at $0.450 \, \mu m$, matching the above results. We could apply these same methods to calculate the average point spacing for HOM-5, obtaining values of $155 \pm 52 \, nm$ and $123 \, nm$ using the Delaunay triangulation and RDF methods, respectively (Supplementary Fig. 42). For striate HOMs, there are no tips on their surfaces. Instead, there are intersection points formed between neighboring plates. In these morphologies, the maximum diameter of accessible particles is limited by the distance between intersection points. We therefore employed the Delaunay triangulation method to calculate the average distance between intersection points, informing a point spacing for HOM-8 of $1.03 \pm 0.322 \, \mu m$ (Fig. 6c).

We conducted a statistical analysis of particle size and point spacing of several HOMs and related materials to showcase the structural diversity of our material (Fig. 6d, e). There is an approximately linear relationship between the particle radius and the point spacing and for most of the particles, the radius is 7.1-fold greater than the point spacing. HOM-5 exhibits the smallest diameter and point

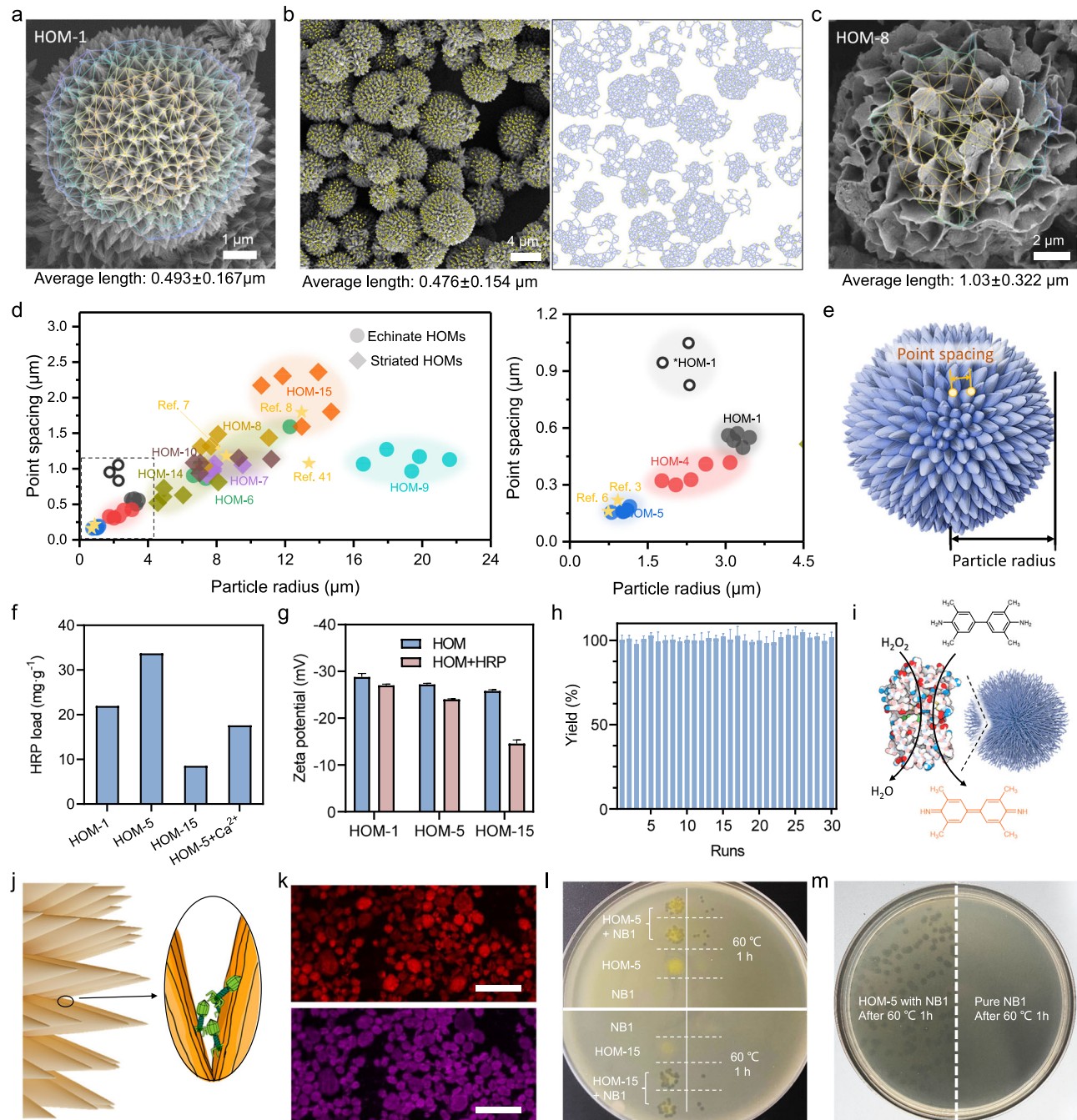

**Fig. 6 | Modeling of the surface morphology and exploration of the applications of HOMs. a** Results of Delaunay triangle partition for a HOM-1 particle. **b** Results of Delaunay triangle partition for multiple HOM-1 particles. **c** Results of Delaunay triangle partition for a HOM-8 particle. **d** Scatter chart of the point spacing versus particle radius for the HOM obtained in this study and the same parameters for some reported microspheres (*: the HOM-1 synthesized in NMP-EtOH). The right figure is an enlarged view of the dashed box in the left figure. **e** Scheme of statistical analyses for HOMs. **f** Zeta potential of three types of HOMs before and after HRP adsorption. **g** Adsorption behavior of HOM-1, HOM-5 (before and after calcination), and HOM-15 in solutions of HRP. Error bars represent standard deviation, *n* = 3. **h** Conversion of TMB in 30 consecutive runs of HRP@HOM-5 catalyzed reactions. Error bars represent standard deviation, *n* = 3. **i** Scheme of HRP immobilized on HOM-5 catalyzing the decomposition of hydrogen peroxide and oxidation of TMB. **j** Scheme of phage immobilized on HOMs. **k** Confocal image for the HOM-15 (top, excited wavelength: 541 nm) and the dye-labelled phages (bottom, excited wavelength: 640 nm). The scale bars are both 40 µm. **l** Lysis plaques of HOMs loaded phage after heat treatment, compared with free phages and pure HOMs. **m** Photo of the HOM-5 loaded phage forming lysis zones on the lawn of *Xoo* after heat treatment, compared with free phages.

spacing, measuring 1.03 µm and 0.161 µm, respectively, while HOM-15 has the largest, measuring 12.9 µm and 2.04 µm, respectively. We analyzed several reported hierarchical assemblies[3,6–8,39] which largely agreed with these trends (marked as stars in Fig. 6d). In the case of HOM-9, significantly larger particle sizes with radii exceeding 21.7 µm are observed alongside 19.3-fold greater point spacing. Furthermore,

HOM-1 particles grown in highly polar solvents (e.g., NMP, EtOH; Fig. 6d) displayed a higher degree of surface fiber entanglement (Supplementary Fig. 43), and the lowest radius-to-point-spacing ratio of 1.9. These quantitative metrics highlighted the morphological diversity of HOM particles, which may be tuned through synthetic conditions and molecular subunits.

## Applications of HOMs in the immobilization of enzymes and phages

The microscale cavities realized in the morphology of HOMs suggested that the immobilization of biomacromolecules and other biological entities might be accessible, in addition to the gas adsorption accommodated by the nanoscale pores present (Supplementary Figs. 8 and 38). The enzyme horseradish peroxidase (HRP) finds applications in wastewater treatment, organic synthesis[40], and immunoassays and diagnostic kits[41], among other biological technologies. The immobilization of HRP and its oxidation of 3,3′,5,5′-tetramethylbenzidine (TMB) was thus assessed with HOMs (Fig. 6d, Supplementary Fig. 44, Supplementary Data 2a). HOM-5 exhibited the shortest point spacing and the highest uptake of HRP (33.7 mg per gram), likely owing the high density of microscale pores. Correspondingly, HOM-15 had the furthest point spacing and the lowest uptake (9 mg per gram). HOM-1 showed intermediate spacing and intermediate uptake (21 mg per gram). The enzyme immobilization capacity of HOM-5 is among the higher ones when compared to traditional inorganic porous materials and polymers (Supplementary Fig. 45, Supplementary Data 2b, Supplementary Table 3). The surface zeta potential of all the HOMs are negative (−5.2 to −29.4 mV) in water, likely due to the presence of carboxyl groups on the surface (Supplementary Fig. 46a). The Zeta potentials of HOM-1, HOM-5, and HOM-15 all increased slightly upon HRP uptake (Fig. 6f, red bar), suggesting electrostatic interactions may further facilitate HRP binding, as HRP carries a positive charge in neutral aqueous solutions (pI = 8.8). After the addition of metal salts, the carboxyl groups on the surface of HOMs are coordinated with metal ions, leading to an increase in their zeta potentials in varying degrees, depending on the types of metals (Supplementary Figs. 46b and 47). As a result, loading capacity of HRP by HOM-5 decreases upon the addition of $CaCl_2$ (Fig. 6g, Supplementary Data 2c), consistent with electrostatic interactions being operative. Pretreatment of HOM-5 with acidic condition, organic solvents and high temperature has almost no impact on its loading capacity of HRP, highlight the robust function of the material under various environmental conditions (Supplementary Fig. 48, Supplementary Data 2d). HOM-5 was selected for recycling studies with the oxidation of TMB, which proceeded to completion even after 30 cycles; these studies further highlight the stability of HOMs even under reactive and oxidizing conditions (Fig. 6h, i).

The uptake of HRP suggested that HOMs with larger cavities might adsorb even larger biological species. Bacteriophages have long been proposed as the next-generation antibacterial agent for clinical and agricultural applications[42]. The shortest point spacing among the HOMs reported herein is around 140 nm, nearly two-fold larger than common Caudovirales phages (~50 nm)[43], offering potential utility in phage immobilization (Fig. 6j). HOM-1, HOM-5, and HOM-15 were thus assessed for phage loading, measured by differences in plaque formation units (PFU) in the supernatant before and after the addition of HOMs. The loading capacity of HOM-1, HOM-5, and HOM-15 were determined to be around $1 \times 10^7$ PFU per mg, suggesting HOMs to be capable of adsorbing Xoo (Xanthomonas oryzae pv. Oryzae) phage (NB1)[44]. A laser scanning confocal microscopy was employed to further characterize the adsorption of bacteriophages onto the surfaces of HOMs. Due to the aggregation-induced emission (AIE) effects, multiple HOM particles are fluorescent, allowing observation under the laser scanning confocal microscopy (Supplementary Figs. 49–51). By staining phages capsid proteins with Cy5[45], the dyed phages could be excited at 640 nm, where pure HOMs were not excited (Fig. 6k, Supplementary Fig. 52). Therefore, when phages were adsorbed on HOMs, NB1@HOMs could be excited at 640 nm. To exclude the influence of dyes adsorbed by HOMs itself, we washed the phages multiple times by ultrafiltration using spin filters (10 kDa mass cutoff). The HOMs immersed in the final filtrate did not exhibit any fluorescence signal under 640 nm excitation, indicating that the aforementioned fluorescence indeed originated from the adsorbed phages. We further found that the HOMs offer a certain degree of protection to NB1, insofar as mitigating their susceptibility to thermal degradation. After treatment at 60 °C for 30 min, the activity of free bacteriophages decreased by 90%, while phages protected with HOM-5 or HOM-15 remained unchanged. In comparison, NB1 protected with classical materials, such as calcium alginate, still showed decreased activity by approximately 70% (Supplementary Fig. 47). When the treatment time was extended to 1 h under 60 °C, phage NB1 entirely lost the lysis ability to infect host Xoo strains. However, in the same condition, both NB1@HOM-5 and NB1@HOM-15 formed lysis plaques on the lawns of their host bacteria Xoo strains (Fig. 6l, m and Supplementary Fig. 44), which indicated that phage NB1 exposed to high temperature still possess antibacterial activity under the safeguard of HOMs.

## Discussion

The hierarchical assembly of porous microspheres from tunable V-shaped molecular building blocks is reported. By modulating the polarity and protic character of solvent during synthesis, we resolve how organic threads form from diverse building blocks through hydrogen bonding and π-π stacking, and how these threads further interact to generate higher-order morphologies including microspheres. We propose this solvents-driven process of self-assembly is general and have developed a series of HOMs based on similar V-shaped molecular subunits. This library afforded HOMs of tunable diameter and pore size, and we introduced two robust mathematical models to describe their surface morphologies, as well as those of reported analogues. These tunable properties allow guest molecules—from several nanometers to several microns in diameter—to be selectively adsorbed, further enhanced through electrostatic interactions, providing a platform for both enzyme and bacteriophage immobilization. Cumulatively, this work expands the design principle of reticular synthesis to include microscopic control over morphology, allowing for diverse structures with distinctive functions from various molecular building blocks.

## Methods

### Preparation of samples

The aldehyde was dissolved in the selected solvent A and mixed with amine in solvent B in equal volumes. The stoichiometric ratio of aldehyde to amine in the reaction is 1:2. After mixing, the solution was left to react for several hours (depending on the choice of substrates), allowing for complete reaction and self-assembly. The precipitate was obtained by centrifugation and washed with solvent B to yield the HOMs. All solvent combinations and reaction times are listed in Supplementary Table 1.

**Nuclear magnetic resonance (NMR).** [1]H and [13]C NMR spectra were recorded on a BioSpin Avance-400 (Bruker, Sweden). Samples were dissolved in DMSO-$d_6$ or 0.1% solution of NaOD in $D_2O$. All spectra were referenced to the proton resonance resulting from the incomplete deuteration of $D_2O$ (δ = 4.79) or DMSO-$d_6$ (δ = 2.50). Solid-state NMR spectra were measured with a 600 MHz Avance III HD NMR spectrometer (Bruker, Sweden).

**Dynamic light scattering (DLS).** The hydrodynamic diameter distributions and zeta potential of HOMs were measured with a dynamic light scattering photometer (ZEN 3600; Malvern Instruments, USA).

**Scanning electron microscopy (SEM).** SEM images were obtained using G300 (Zeiss, Germany) and with a secondary electron detector, operating at 3 kV using a Schottky thermionic emission electron gun. High-resolution SEM images were acquired by using field-emission SU-8010 microscopy (Hitachi, Japan) with a secondary electron detector. Before imaging, the samples were sputtered with gold (nano-sized

film) for 60 s using a HITACHI MCI000 ion sputter to avoid charging during SEM analyses. The samples were prepared simply by putting a drop (about 1 μL) of dispersed samples in volatile solvents on clean aluminium foil.

**Cryo-electron microscopy (Cryo-EM).** All samples were suspended in water (about 17 mM in concentration) and processed with ultra-sonication for 30 min. The samples were applied to a holey carbon grid covered with graphene-oxide (Quantifoil R1.2/1.3, Au, 300 mesh). After 3 s, the grids were blotted for 3.5 s at a humidity of 100% and temperature of 20 °C and then plunge-frozen in liquid ethane using a Vitrobot (FEI, Netherlands). The images were collected using a Talos F200C 200kv microscope (FEI, Netherlands).

**High-resolution Mass Spectra (HRMS).** HRMS were obtained on an Agilent 6545 quadrupole-time of flight mass spectrometry with electrospray ionization (Agilent, USA). All the samples were dissolved in DMSO, or for insoluble materials, in 14 mM ammonia solution in water. The solutions were filtered with a nylon membrane with a pore size of 0.22 μm before analysis.

**Powder X-ray diffraction (PXRD).** PXRD patterns were recorded on a D/Max-2550pc (Rigaku, Japan), using Cu Kα radiation (tube operating at 40 kV and 250 mA) with a scintillation detector. Synchrotron radiation XRD is collected at the National Synchrotron Radiation Laboratory at the University of Science and Technology of China with a wavelength of 0.67092 A.

**Laser scanning confocal microscopy.** The confocal images were obtained using a CLSM600 laser scanning confocal imaging system equipped with an IRX60 inverted microscope (SOPTOP, China).

**Single-crystal X-ray determination.** SCXRD data were collected using a D8 venture X-ray diffractometer (Bruker, Sweden) equipped with METALJET liquid metal (Ga) X-ray source.

**Thermogravimetric analysis.** TGA was carried out on a TA SDT Q600 under $N_2$ atmosphere from room temperature (around 28 °C) to 710 °C along with a ramp rate of 10 °C min$^{-1}$.

### MD simulation

Unless specified otherwise, molecular models were optimized by DFT calculations using M06-2X functional with 6-311 + G(d,p) basis set for all the atoms in Gaussian 16 software package[46,47]. MD simulation was performed using the GROMACS 2020.1 package using Generation Amber Force Field (GAFF). A pre-ordered model or 100 randomly dispersed molecules in a periodic box of $10 \times 10 \times 10$ nm$^3$ containing a mixture of DMSO and MeOH (2978 molecules for each) were pre-treated with energy minimization, then simulated for 100 ns, respectively, in the NPT ensemble with time steps of 1 fs. Bonds involving hydrogens were constrained using the LINCS algorithm. Temperature was kept at 298 K using the velocity-rescaling thermostat ($\tau_T = 1.0$ ps) and pressure at 1.0 bar using the Berendsen barostat ($\tau_p = 1.5$ ps). Van der Waals and electrostatic forces were cut off at 1.4 nm using the Verlet list scheme; long-range electrostatic interactions were treated using a Barker–Watts reaction field with $\varepsilon_{RF} = 62$. Visualization of the simulations was done using the VMD program v. 1.9.3. The dihedral angles of BACT after MD simulations were calculated using Supplementary Code 1.

### Establishment of coordinate systems and surface modeling of HOMs

The coordinates for the tips of the echinate HOMs and the intersection points of the striated HOMs were generated using ImageJ software from the corresponding SEM images. By default, the pixel at the top left corner of the SEM image was set as the origin point. We introduced a third z-coordinate to generate three-dimensional coordinates of all the points from their two-dimensional projections in SEM. An ellipsoid model with the following general formula.

$$\frac{(x-a)^2}{A} + \frac{(y-b)^2}{B} + \frac{z^2}{C} = 1 \tag{1}$$

Here, $(x, y, z)$ is the coordinate of a point. $(a, b, 0)$ is the center coordinates of the ellipsoid. A, B, and C are the major axis, intermediate axis, and minor axis of the ellipsoid, respectively.

Subsequently, based on the corresponding surface equation, we deduce and calculate the z-axis coordinate for each point (Supplementary Methods 8 and Supplementary Code 2). For the images with multiple particles, the recognition of tips was automatically performed by ImageJ. Only tips from microspheres at the top layer were considered. Undesired tips for microspheres at deeper layers can be filtered by adjusting the brightness threshold since the tips at the top tend to appear brighter due to discharge effects in electron microscopy.

### Geometric parameterization of HOMs based on the Delaunay triangulation

After obtaining the coordinates of selected points, we performed Delaunay triangulation on the point sets using MATLAB R2021a (Supplementary Methods 9 and Supplementary Code 3). We employed the built-in Delaunay module in MATLAB to perform triangulation to obtain the corresponding point connection. Subsequently, taking into account the z-axis coordinates, we calculated the average length of all triangle edges, which was set as the initial average. Based on the closest packing model, the distance from a tip to its second nearest tip is √3 times the distance to the nearest tip. So, the threshold value will be set as √3 times the initial average. All the larger lengths will be filtered. Then, we recalculated the average of the remaining lengths as the final result. The overall procedure is illustrated in Supplementary Figs. 19 and 20, and all the related data is listed in Supplementary Data 1.

### Geometric parameterization of HOMs based on the radial distribution function (RDF)

In this algorithm, we measured the distances between every pair of points in the point set (marked as $D$) regardless of whether they were neighboring. Distances larger than max, set as the diameter of the largest particle, are omitted to reduce the size of the dataset. We tallied how many distances fell within a circular ring. This ring had an inner distance, r, and a width of $\Delta r$. This count is denoted as $N(r)$, representing the number of distances where $r \leq D \leq r + \Delta r$. Additionally, we tallied the total number of distances which have been considered, labelled as $N$. Then, using the obtained data, we calculated the radial distribution function, denoted as $g(r)$, using the following formula. The peaks in the radial distribution function indicate that for a given point in the given point set, it is most likely to find another point at that particular distance, or in other words, a neighboring point (Supplementary Methods 10 and Supplementary Code 4).

$$g(r) = \frac{N(r) \bullet r_{max}^2}{N \bullet (2r \triangle r + \triangle r^2)} \tag{2}$$

### Catalyst recycles of HOM-immobilized HRP

50 μg of HRP was immobilized on 2 mg of HOM-5, followed by the addition of 150 μL of TMB liquid substrate solution (TMB concentration: 200 μg/mL) diluted to one-twentieth of its original concentration. TMB liquid substrate solution contained both TMB and peroxide as the oxidizing reagent. After the reaction at 25 °C for 1 min, the HRP@HOM-

5 material was separated from the reaction mixture by centrifugation at 9569 x g at 4 °C for 2 min. The absorbance of the supernatant at 450 nm was measured and compared with the absorbance of the reaction solution catalyzed by non-immobilized HRP under the same conditions. The recycled HOMs were then added to 150 μL of diluted TMB liquid substrate solution for the next round of reaction.

### Determination of the thermotolerance of HOMs-loaded bacteriophages

Bacteriophage NB1 was isolated following the reported method with slight modifications[44]. *Xanthomonas oryzae* pv. *oryzae* strain N1 was used as the phage host. Phage stock was adjusted to a titer of $1 \times 10^9$ plaque-forming units per milligram (PFU/mL). Initially, bacteriophages NB1 and the HOM-5 or HOM-15 suspension (10 mg/mL) were mixed in a volume ratio (10:1) and subjected to overnight shaking at 10 °C. Subsequently, high-speed centrifugation at 11,000 $g$ for 10 min was performed to facilitate the separation of HOMs loaded with bacteriophage NB1, resulting in their precipitation at the bottom of the tube. The supernatant, containing non-adsorbed bacteriophages, was meticulously transfered using pipette tips. Subsequently, the supernatant was subjected to filtration through a 0.22 μm syringe filter to eliminate any residual impurities. The titer of unadsorbed free phage present in the clarified supernatant was determined using the double-layer method, a widely employed technique for quantification. Based on the quantified titer of unadsorbed free phage, it was ascertained that both HOM-5 and HOM-15 demonstrated an adsorption efficiency of $10^7$ PFU/mg.

The thermal stability of bacteriophage NB1 was assessed by incubating the phage at a range of temperatures. Phage NB1 was subjected to thermal treatments at temperatures of 50 °C, 60 °C, 70 °C, and 80 °C for durations of 30 min and 60 min, respectively. The experimental results revealed that after incubation at 60 °C for 60 min, phage NB1 completely lost its ability to lyse and infect the *Xoo* strains, indicating a significant loss in infectivity. Subsequently, the spot-test method was employed to assess the activity of both the free phage and the phage that had undergone heat treatment. The bacterial strain *Xanthomonas oryzae pv. oryzae* N1, which serves as the host for the phage, was cultivated and combined with molten semi-solid NA medium to generate double-layer plates. The phages in different treatments were 10-fold diluted with ddH2O, 2 μL of each phage titer was spotted on the surface of the plates. After incubating at 30 °C overnight, phage activity could be assessed through the plaque formation. Meanwhile, we also spotted the HOM-5 and HOM-15 without phage on the plate to see whether the materials themselves would damage the bacteria *Xoo* N1. To demonstrate bacteriophage NB1 activity more intuitively, we mixed 5 ml of NA semi-solid broth (0.8% agar), 1 mL of host bacteria *Xoo* N1 suspension, and 100 μL of diluted wild phage or phage adsorbed on HOM-5 and HOM-15 evenly, and poured them together onto NA plates. The double-layer plates were placed overnight at 30 °C for cultivation. The appearance of uniform plaques on the double-layer plates indicated the phage activity.

### Reporting summary

Further information on research design is available in the Nature Portfolio Reporting Summary linked to this article.

## Data availability

The crystallographic data generated in this study have been deposited in the Cambridge Crystallographic Data Centre (CCDC) referencing deposition no. 2310928 [https://doi.org/10.5517/ccdc.csd.cc2hkq2w]. The dataset generated and analysed during the current study is available at [https://doi.org/10.5281/zenodo.11334183]. The data is available from the authors on request.

## Code availability

The codes and datasets in this study have been deposited in the Zenodo database under Creative Commons Attribution 4.0 International Public License at [https://doi.org/10.5281/zenodo.11335121].

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

## Acknowledgements

We thank S. Chang and L. Wu for assistance with cryo-TEM at the Center of Cryo-Electron Microscopy (CCEM), Zhejiang University. We thank the Chemistry Instrumentation Center of Zhejiang University for instrumental support, including F. Chen for assistance with scanning electron microscopy; Y. Liu and M. Yu for assistance with NMR; Q. He for help with MS; and X. Hu for XRD analysis. We thank the Bio-ultrastructure Analysis laboratory of the Analysis Center of Agrobiology and Environmental Sciences of Zhejiang University for instrumental support, including N. Rong and J. Niu for assistance with scanning electron microscopy. The authors would like to thank Shiyanjia Lab (www.shiyanjia.com) for the Single-crystal and fluorescence spectra analysis. We thank Dr. D. Dong at the Oak Ridge National Lab for helping with performing DFT and MD calculations. This work was supported by NSFC (22172142, P.J. and 22377107, P.J.), Zhejiang Provincial Outstanding Youth Science Foundation (LR22B030004, P.J.), and Fundamental Research Funds for the Central Universities (226-2024-00003, P.J.).

## Author contributions

P.J. and Y.L. initiated this project. Y.L. synthesized all samples and performed most examinations. L.F. helped perform PXRD and analysed the results. X.X. and B.L. finished the experiments on phages and analysed the results. Y.S. and W.W. helped perform catalyst recycles. S.V. analysed the kinetics data. Y.L., S.V., and P.J. wrote the manuscript. All authors have reviewed and approved the manuscript.

## Competing interests

The authors declare no competing interests.
