## [Peer Review File · Nature Communications]

Hierarchical Organic Microspheres from Diverse Molecular Building BlocksREVIEWER COMMENTS

Reviewer #1 (Remarks to the Author):

The authors describe various spherical objects formed from V-shaped carboxylic acid derivatives. However, the reviewer thinks that there are leaps in theory.

The Introduction mentions many porous frameworks (MOF, COF, and especially HOF), while the macrosphere mentioned in the Results and discussion does not seem to be related to porous frameworks. There is no consistency between the introduction and the text. Furthermore, there is no evidence of framework structure in the microsphere except for MD calculation models.

The authors describe that Fig. 1 shows the design and synthesis of hierarchical organic microsphere. It is, however, more like a hierarchical interpretation or a hierarchical scenario, not a design guideline. In particular, the process of forming a spherical assembly from a thread or sheet-like assembly is not inevitable, and there is no element of design at all. This diagram does not show “design”. In addition, the cartoon of the molecule may be intended as a hydrogen bonding site, but it is also unclear how the hydrogen bond functions to give the ring shape and head-to-tail alignment. Formation schemes of COFs and HOFs shown in Fig. 1a,b has basically no relation to Fig. 1c. Consequently, the reviewer does not recommend publication in Nature Communications. A logical structure in the manuscript should be reconstructed.

Reviewer #2 (Remarks to the Author):

This manuscript describes interesting experimental results on synthesis of hierarchical organic microspheres (HOMs) synthesized from V-shaped molecular building blocks through hydrogen bonding and π - π stacking. The research highlights the influence of solvent polarity and protic character on the assembly process and introduces mathematical models for describing surface morphologies. Applications demonstrated include enzyme immobilization for catalytic reactions and bacteriophage adsorption for potential antibacterial use. This work is very interesting. The following concerns should be addressed properly before considering this work for publication in Nature Communications.

1. While the paper provides detailed insights into how the selection of solvent and the interplay of non-covalent interactions, specifically hydrogen bonding and π - π stacking, can be leveraged to control the self-assembly of molecular building blocks into microspheres with diverse morphologies and functions, it appears to lack comprehensive data on the growth, characteristics, and properties of each particle type.

2 While the paper details the final shapes of particles after the reaction has completed, it does not provide data on the time-dependent growth behavior of the particles. Time-lapse studies could offer insights into the nucleation and growth mechanisms, as well as the evolution of particle

morphology.

3. The manuscript does not specifically address the effects of temperature during the synthesis process, which suggests that a detailed investigation into how temperature influences the assembly and morphology of the hierarchical organic microspheres is lacking. The effects of temperature during the synthesis process should be described in details.

4. The manuscript does not provide comprehensive validation or detailed analysis regarding the long-term stability of the particles and their functionality across diverse environmental conditions. Such studies would be crucial for assessing the practical applicability of these microspheres in real-world applications, where stability and functional integrity over time and in different environmental matrices are paramount.

5. The manuscript does not sufficiently integrate detailed explanations for the results shown in the Supplementary Figures, including the FT-IR results (Supplementary Fig. 2) and the thermal stability results from TGA (Supplementary Fig. 3), into the main text. In addition, while zeta potential value is included in the supporting figures, it is crucial that these values are also discussed in detail within the main text of the document.

6. The paper indeed discusses the applications of hierarchical organic microspheres (HOMs) in the immobilization of enzymes and bacteriophages but lacks a quantitative or qualitative comparison with systems reported in existing literature.

7. The sentence on page 1, "Polystyrene microparticles, for example, are common platforms for immunoprecipitation and related techniques," along with its reference to ref. 10, seems out of place within the document. Since ref. 10 does not concern polystyrene microparticles, removing this sentence would help maintain the coherence and relevance of the discussion.

Reviewer #3 (Remarks to the Author):

Ji and co-workers present a significant work, altering the starting molecular building blocks and conditions to produce a wide variety of microspherical architectures. I have been asked to comment particularly on the computational aspects of the work and so I focus my comments here. As most of the computational detail is contained in the SI, I have read the SI in more detail than the manuscript itself.

I think the key sentence of the authors' work is written on Line 66-68, that "... the self-assembly proceeds through threads or sheets, depending on the relative strength and directionality of pi-pi stacking and hydrogen bonding."

I agree with this statement from the authors and largely my comments center about how well they

have addressed this proposal.

My overall impression is that the authors have been both careful and thorough in the series of calculations they have done, but they haven't spent the same time on pi-pi stacking as they have hydrogen bonding. I appreciate the code provided in the SI, but I do note that it is difficult to use the code as given. It would be beneficial if the code was additionally uploaded to (e.g.) github with enough sample data to both check the code, and the results in the paper.

My largest criticism of the computational work is that it seems to be entirely focussed on hydrogen bonding. Supplementary method 4 and the figures 4-6 cited therein, focus only on hydrogen bonds. I don't see any pi-pi stacking to compare to. I think the authors could, relatively easily, cut from their MD simulation a selection of (partly) overlapping dimers and produce something similar to SI Fig 6, but for pi-pi stacking.

Smaller points:

Manuscript line 400 and in the SI, Page 3, a 2 fs timestep is cited. This seems to me quite large, normally the largest timestep ensuring energy conservation would be 1 fs.

SI, Page 4, middle paragraph: The enumeration of the different intermolecular hydrogen bonds is appreciated and sets out a sensible calculation strategy. I think though, at the end of this paragraph, there might be an autocorrect - I think the authors mean "*inter*molecular interactions between two molecules".

SI Fig 5: It wasn't quite clear to me whether the two dihedral angles highlighted were independent (I think so) or linked. Assuming they were independent, were their distributions equivalent?

SI Fig 6 c) and d): The dotted lines indicating the intermolecular interactions were slightly difficult to see - highlighting them in a brigher color would help a lot.

Typographical errors:

Manuscript Line 74/75: "With respect to function, we applied these sixteen HOMs displayed..." I think two sentences got mixed together here. If the words "we applied" were deleted, I think it would say what the authors mean.

SI Page 5: "spaces group" -> space group
"force flied" -> force field

REVIEWER COMMENTS

Reviewer #1 (Remarks to the Author):

The authors describe various spherical objects formed from V-shaped carboxylic acid derivatives. However, the reviewer thinks that there are leaps in theory.

The Introduction mentions many porous frameworks (MOF, COF, and especially HOF), while the macrosphere mentioned in the Results and discussion does not seem to be related to porous frameworks. There is no consistency between the introduction and the text.

Response: We thank the reviewer for the insightful comments and the time dedicated to reviewing our work.

We understand the concern from the reviewer that the introduction have overly emphasized on framework materials. We have restructured the second paragraph of the introduction and Figure 1. We have removed any discussion over MOFs since they are less related to our materials which contain no metal-based building units. We modified our discussion over COFs and HOFs to focus on their properties as modularly synthesized porous organic materials, which is the exact property of HOMs, that are modularly synthesized from various dialdehydes and amino carboxylic acids, assembled from V-shaped organic building blocks, and the materials are uniquely porous for their micron-sized cavities. We believe that the modified introduction can provide a more accurate presentation of the potential and versatility of modularly synthesized porous organic materials, and better highlight HOMs as a new group of materials in this category.

Furthermore, there is no evidence of framework structure in the microsphere except for MD calculation models.

Response: We thank the reviewer for the suggestions. We have added CO₂ sorption isotherm during the revision, and now we have three aspects of evidence on the framework structure of the microsphere.

First, CO₂ is a commonly used sorbent molecule for analyzing sub-nanoscale sized pores, as confirmed and extensively studied in previous reports (Microporous Mesoporous Mater.2016, 224, 294-301; J. Mater. Chem. A.2017, 5, 25014-25024; Small Methods.2018, 2, 1800173). Therefore, we tested the CO₂ sorption isotherms of HOM-1 at 273K to more accurately measure the presence of sub-nanoscale sized pores. The sizes of measured pores are 5.6 Å and 8.0 Å (Supplementary Fig. 8b-c).

Second, MD calculations were used to simulate the hollow tubular structure which matched well to the PXRD diffraction patterns, also showed the presence of two types

of sub-nanoscale pores with sizes of 5.7 Å and 8.2 Å (Supplementary Fig. 8d), corresponding well to the results from CO₂ sorption isotherm.

Third, in PXRD, the diffraction signal peak at 5.44°, corresponding to (110) plane, gives a calculated distance of 1.67 nm. Which matches to the half size of the simulated hollow tubular structure packed from the tetrameric ring structure. The fringes in high-resolution TEM has a measured distance of 1.64 nm, further validated the presence of hollow square tubular structure. Therefore, we believe that the framework structure exists in HOM-1.

Regardless, framework structure is not the key aspect of this work. As highlighted in our title “Hierarchical Organic Microspheres from Diverse Molecular Building Blocks”, the major discovery of this work is the synthetic strategy to access diverse organic microspherical materials. The pores in the hollow tubular structure is not the major structural feature. The larger cavities formed through the assembly of the primary tubular or planner structure are the major focus of this work, and were demonstrated for loading proteins as well as larger-sized bacteriophages. This work is a report of the modular synthesis of diverse microspheres from dialdehydes and amino carboxylic acids.

Supplementary Fig. 8 | **a**, N₂ sorption isotherms at 77 K of HOM-1 the precipitate of triple condensation product (TACT). From the isotherm, the surface area (S_{BET}) of the HOM-1 and TACT precipitate was calculated to be 55.8 m² g⁻¹ and 3.0 m² g⁻¹, respectively. Few pores in the TACT may have already deviated from the BET model. **b**, CO₂ adsorption–desorption isotherms at 273 K of HOM-1. **c**, Calculated pore size distribution plot of HOM-1 from CO₂ adsorption data at 273 K after DFT model

fitting of adsorption branch data. **d**, Schematic representation of the two main kinds of pore in the HOM-1 model.

The authors describe that Fig. 1 shows the design and synthesis of hierarchical organic microsphere. It is, however, more like a hierarchical interpretation or a hierarchical scenario, not a design guideline. In particular, the process of forming a spherical assembly from a thread or sheet-like assembly is not inevitable, and there is no element of design at all. This diagram does not show “design”.

Response: We thank the reviewer for the important comment. We agree that the term “design” is not accurate. We designed the 16 types of V-shaped building units, and we recognized a pattern that all these molecules have assembled into hierarchical microspheres. The accurate assembly modes of these V-shaped monomers were not designable. Therefore, we revised the figure caption for Fig. 1 from “design and synthesis of hierarchical organic microsphere” to “Structural feature of hierarchical organic microspheres in comparison with classical porous organic materials”. The caption for Fig. 1c is changed to “design of V-shaped molecular building blocks and the proposed self-assembly mechanism to form hierarchical organic microspheres”.

In addition, the cartoon of the molecule may be intended as a hydrogen bonding site, but it is also unclear how the hydrogen bond functions to give the ring shape and head-to-tail alignment.

Response: We thank the reviewer for the query. We have added chemical structures to more clearly show the intermolecular interactions that formed the ring structure and head-to-tail alignment, as shown below:

The ring shape model used for explaining the structure of HOM-1 was proposed to form through the hydrogen bonding between the carboxyl groups with the phenol-imine pair. We have provided a detailed discussion of the hydrogen bonding assembly

mode in HOM-1 in Supplementary Method 4. Essentially, we used DFT to compare 7 types of possible intermolecular interactions. The carboxyl group to phenol-imine interaction gave the lowest total binding energy. Based on the PXRD data, the simulated annealing method was performed to create initial assembly model, followed by geometry optimization. We extracted this cyclic model from a single-layer crystal cell (supplementary method 7 and Fig.2j).

The head-to-tail alignment was proposed to form through the hydrogen bonding interactions between the carboxyl group and the polar aryl C-H bonds of furan and pyridine rings. The intermolecular interaction between the two imine groups and the furan polar C-H bonds was observed in the single crystal of **18** with DMAc (CCDC code: 2310928), as shown below:

Formation schemes of COFs and HOFs shown in Fig. 1a,b has basically no relation to Fig. 1c.

Response: We have redrawn the formation scheme in Fig. 1 to better present the structural relation and similar synthetic modularity of HOMs to the synthesis of COFs and HOFs. We think it is important to compare HOMs with classical porous organic materials since they share the same synthetic principle, which the use of small organic building blocks and their bonding/intermolecular interactions to generate macroscopic functional structures through modular synthesis.

We hope that the revised Figure 1 could better show the relation of Fig. 1a, b to Fig. 1c at least from two aspects. First, from the standpoint of synthetic strategy, HOMs are symmetry-reduced version of HOF materials, from highly symmetric building blocks (such as D_{3h}-symmetry tricarboxylic acids) to V-shaped building blocks, thus leading to more sophisticated hierarchical assembly with microspherical structure. Second, from the standpoint of intermolecular interactions, COFs were formed from aldehyde linkers and amine linkers through the covalent interactions, which is similar to HOMs for which the building blocks were formed from imine condensation of aldehydes and amines. HOFs were formed through the hydrogen bonding of carboxyl

groups and π - π stacking, which are also the major modes of intermolecular interactions for forming the hierarchical assembly in HOMs.

Consequently, the reviewer does not recommend publication in Nature Communications. A logical structure in the manuscript should be reconstructed.

Response: We thank the reviewer for the critical assessment of the introduction section of our work, and the important constructive suggestions. We have reorganized the logical structure and made adjustments to the introduction and Figure 1. We hope that these modifications can provide a more accurate presentation of the potential and versatility of modularly synthesized porous organic materials, and better highlight HOMs as a new group of materials in this category.

Reviewer #2 (Remarks to the Author):

This manuscript describes interesting experimental results on synthesis of hierarchical organic microspheres (HOMs) synthesized from V-shaped molecular building blocks through hydrogen bonding and π - π stacking. The research highlights the influence of solvent polarity and protic character on the assembly process and introduces mathematical models for describing surface morphologies. Applications demonstrated include enzyme immobilization for catalytic reactions and bacteriophage adsorption for potential antibacterial use. This work is very interesting.

Response: We thank the reviewer for the thorough and comprehensive review, and for the endorsement of our work.

The following concerns should be addressed properly before considering this work for publication in Nature Communications.

1. While the paper provides detailed insights into how the selection of solvent and the interplay of non-covalent interactions, specifically hydrogen bonding and π - π stacking, can be leveraged to control the self-assembly of molecular building blocks into microspheres with diverse morphologies and functions, it appears to lack comprehensive data on the growth, characteristics, and properties of each particle type.

Response: We thank the reviewer for the constructive suggestions. We have included additional data to fully characterize each of the 16 HOM materials as suggested. For the growth of the materials, we monitored the growth rate of each HOM *in situ* through their turbidity at 600 nm. The data were included in the Fig. 3 and supplementary Fig. S18-32. For all the 16 types of HOMs, we characterized their chemical composition through ^1H NMR and high-resolution mass spectrometry, as well as their crystallinity through PXRD (Fig.2 and supplementary Fig. 18-32).

For the properties of the materials, we reported the fluorescence spectra (supplementary Fig. 18-32), along with their illuminated images of the materials (supplementary 50), which showed that most HOMs exhibited strong fluorescence. The porosity of the materials was characterized with N_2 adsorption isotherms, with their calculated BET surface areas summarized in Supplementary Fig. 38. We further analyzed the surface charge properties of each material through zeta potential (supplementary Fig. 46).

2 While the paper details the final shapes of particles after the reaction has completed, it does not provide data on the time-dependent growth behavior of the particles. Time-lapse studies could offer insights into the nucleation and growth mechanisms, as well as the evolution of particle morphology.

Response: We thank the reviewer for the suggested experiment. We performed time-lapse studies by centrifuging assembly intermediates of HOM-1 at different time

points followed by SEM imaging. At 90 seconds, small clusters were formed through the entanglement of short fibers. At 120 seconds, these clusters started to exhibit branching structures at both ends. And at 150 seconds, the branching structure at the ends became more pronounced, along with the emergence of spherical particles. These results were included in Supplementary Fig. 10, which indeed offer insights into the nucleation and growth mechanisms of HOM material.

Supplementary Fig. 10 SEM image of intermediates isolated at 90s (a-b), 120s (c-d), 150s (e-f).

3. The manuscript does not specifically address the effects of temperature during the synthesis process, which suggests that a detailed investigation into how temperature influences the assembly and morphology of the hierarchical organic microspheres is lacking. The effects of temperature during the synthesis process should be described in details.

Response: We thank the reviewer for the important suggestion. We tested and confirmed the significant influence of temperature on the molecular self-assembly process.

We monitored the dynamics of the assembly process of HOM-1 at 5, 25 and 45 °C using turbidity measurements, as shown in Supplementary Fig. 16a. There was an induction period for each of the three curves, featured by no increase in turbidity in the first few minutes, likely because that this stage is primarily dominated by the imine condensation reaction to yield the doubly condensed product. At 0°C, the

induction period lasted for 4 minutes. As the reaction temperature increases, the duration of this process gradually shortens to 2.5 minutes at 25 °C and 2 minutes at 45 °C. In the assembly stage, the turbidity of the suspension increased rapidly. The rate of the assembly was analyzed by taking the first derivative of turbidity (Supplementary Fig. 16b). At 5 °C, the maximum rate of the assembly is higher than those of 25 and 45 °C, likely because that the building blocks are less soluble at lower temperature.

We further studied how temperature influenced the morphology by synthesizing all the 16 materials under varied temperatures, followed by SEM imaging. Take HOM-1 for example, the material was synthesized at 0, 25 and 50°C. At 0 °C, the material showed uneven particle sizes ranging from 0.8 to 5.7 nm, which may be caused by more rapid assembly (Supplementary Fig. 16c). Conversely, at 50 °C, the particle size was much larger (about 11 nm) than at 25 °C. Overall, for HOM-1, room temperature remains optimal for synthesis. All other HOM materials were also synthesized at 0, 25 and 50°C, with their SEM images shown in supplementary Fig. 17. In conclusion, temperature has significant influence on both the assembly dynamics and morphology of the HOMs, and is a valuable parameter to tune for accessing varied HOM materials.

Supplementary Fig. 16 a, Effect of temperature on precipitation kinetics. **b**, The first order differential of turbidity recorded under different temperature. **c**, SEM image of HOM-1 synthesized under different temperature (the scale bar is 3 μ m).

Supplementary Fig. 17 | a-o, SEM image of HOM-2~16 synthesized under 0°C or 50 °C.

4. The manuscript does not provide comprehensive validation or detailed analysis regarding the long-term stability of the particles and their functionality across diverse environmental conditions. Such studies would be crucial for assessing the practical applicability of these microspheres in real-world applications, where stability and functional integrity over time and in different environmental matrices are paramount.

Response: To assess the long-term stability of the material, we took SEM images of all the 16 materials synthesized around one year ago (on May 7 of 2023), which showed no significant difference in their morphology compared to freshly synthesized materials (Supplementary Fig 34).

To assess the stability of the material under diverse environmental conditions, we immersed all HOM materials in harsh pH conditions followed by turbidity measurement and SEM imaging. Under acidic conditions with pH of 1, all the 16 materials exhibited high stability as measured through their unchanged turbidity and morphology (Supplementary Fig 36, 37). Under alkaline conditions (pH=12), HOM-9, 13, and 15 remained stable evidenced by no decrease in turbidity. Moreover, surface metallization can be employed to enhance the stability of HOM under alkaline conditions. As shown in supplementary Figure 47, HOM-1 completely dissolved in a phosphate buffer at pH=7.5. But after the addition of Ca^{2+} or Mg^{2+} , it remained undissolved in a buffered solution even at pH=10.

To test the thermostability of the materials, we recorded the thermogravimetric curves for each material, to find that that the decomposition temperatures of all the HOMs are above 240°C (supplementary Fig 35).

To further investigate the functionality of the materials in different environmental matrices, we selected the system of using HOM-5 for immobilization of HRP. We treated HOM-5 under acidic conditions, basic conditions, high temperature and organic solvent, followed by measurement of enzyme loading capacity. As shown in supplementary Fig. 48, after treatment under different conditions for 14 hours, HOM-5 exhibited no significant difference in adsorption capacity for HRP, with maintained loadings capacity of over 95% in most group. This indicated that HOM-5 maintains good functional stability under diverse environmental conditions.

Supplementary Fig. 34 | a-p, SEM image of HOM-1~16 samples after being stored in powder form at ambient condition for 1 year.

Supplementary Fig. 36 | Relatively turbidity of HOMs in HCl or NaOH solution. 1 mg HOMs were dispersed in 100 μL H_2O . 20 μL HOMs suspension was added into 180 μL HCl (pH = 1) or NaOH (pH=12) solution. The OD_{600} was measured as sample turbidity after 10 min vortex. Also, the reference turbidity was measured with HOMs dispersed in pure H_2O (1 mg mL^{-1} , 200 μL). The relative turbidity is the ratio of the sample turbidity in an HCl or NaOH solution to the reference turbidity.

Supplementary Fig. 37 | **a-p**, SEM image of HOM-1~16 samples after being immersed in HCl solution (pH=1) for 12 hour.

Supplementary Fig. 35 | a-p, Left: Thermogravimetric analysis (TGA) of the HOM-1~16. **Right:** The first derivative of the TGA curve.

Supplementary Fig. 47 | **a**, The retention ratio curves of HOM-1 with or without metal ions, which were recorded by the absorbance of supernatant. **b**, The SEM image of HOM-1 treated with Ca²⁺. **c**, The SEM image of HOM-1 treated with Mg²⁺. The scale bars for (c) and (d) are both 1 μm.

Supplementary Fig. 48 | a, Residual HRP loading capacity of HOM-5 after treatment under different conditions. HOM-5 was suspended in solvents, prepared as a 10 mg/mL suspension. After 14 hours, the original solvent was removed by centrifugation, and HOM-5 was washed three times with deionized water, and ultimately suspended in deionized water for testing its adsorption capacity for HRP. The “50°C” group refers to heating the HOM-5 powder at 50°C for 14 hours, dispersing it in deionized water, and testing its adsorption capacity. **b,** The SDS-PAGE image of protein supernatant without HOM-5 (the first lane) and with HOM-5 after treatment under different conditions (the 2nd to the 9th lanes).

5. The manuscript does not sufficiently integrate detailed explanations for the results shown in the Supplementary Figures, including the FT-IR results (Supplementary Fig. 2) and the thermal stability results from TGA (Supplementary Fig. 3), into the main text.

Response: We have integrated the discussion over FT-IR and TGA into the manuscript as following:

“The condensation was also confirmed by the disappearance of C=O stretching in the FT-IR spectrum of HOM-1 (Supplementary Fig. 2).”

“All the 16 HOMs showed high stability after long-term storage for one year (Supplementary Fig 34), and thermal treatment, with decomposition temperatures of above 240 °C (Supplementary Fig 35).”

In addition, while zeta potential value is included in the supporting figures, it is crucial that these values are also discussed in detail within the main text of the document.

Response: We have integrated the discussion over zeta potential into the main text as following:

“The enzyme immobilization capacity of HOM-5 is among the higher ones when compared to traditional inorganic porous materials and polymers (Supplementary Fig. 45, Supplementary Table 3). The surface zeta potential of all the HOMs are negative (-5.2~-29.4 mV in water, likely due to the presence of carboxyl groups on the surface (Supplementary Fig 46a). The zeta potentials of HOM-1, HOM-5, and HOM-15 all increased slightly upon HRP uptake (red bar in Fig. 6f), suggesting electrostatic interactions may further facilitate HRP binding, as HRP carries a positive charge in neutral aqueous solutions (pI = 8.8). After the addition of metal salts, the carboxyl groups on the surface of HOMs are coordinated with metal ions, leading to an increase in their zeta potentials in varying degrees, depending on the types of metals (supplementary Fig 46b and 47).”

6. The paper indeed discusses the applications of hierarchical organic microspheres (HOMs) in the immobilization of enzymes and bacteriophages but lacks a quantitative or qualitative comparison with systems reported in existing literature.

Response:

For the immobilization of enzymes, we test several classic materials for HRP loading, including celite (Bioresour. Technol.2007, 98, 1012-1019), activated carbon (Bioresour. Technol.2016, 210, 108-116; Sci. Bull.2004, 49, 2452-2454), chitosan (Food Hydrocoll.2023, 139, 108551), and silica (Chem. Soc. Rev.2013, 42, 6277-

6289). Among the classic materials tested (supplementary Fig. 45), silica exhibited the highest loading capacity for HRP of 13.1 mg/g, much lower than the loading capacity of HOM-5 (33.7 mg/g). Compared to the tested traditional inorganic porous materials and organic polymers, HOM-5 exhibits a certain advantage in enzyme immobilization capacity.

Carrier	Loading Capacity	Reference
Chitosan–halloysite hybrid-nanotubes	21.5 mg/g	Chem. Eng. J. 2013, 214, 304-309
poly(GMA-MMA)	3.35 mg/g	J. Hazard. Mater. 2008, 156, 148-155.
Magnetic biochar	65 mg/g	J. Hazard. Mater. 2020, 384, 121272.
SOM-ZIF-8	71.2 mg/g	Inorg. Chem. Front. 2020, 7, 3146-3153
Calcium alginate	8.9 mg/g	Polymers. 2022, 14, 2614.
Silica nanoparticles	8.06 mg/g	Colloids Surf. B. 2023, 229, 113443.
poly (Pro-Glu) modified silica gel	16.8 mg/g	Talanta. 2022, 241, 123223.
multi-walled-carbon-nanotube/cordierite composite	1.34 mg/g	Process Saf. Environ. Prot.2017, 107, 463-467.
HOM-5	33.7 mg/g	This work

Table R1. Comparison of HRP loading capacity in this work to other reported carriers.

Supplementary Fig. 45 | a, Loading capacity of HOM-5 for HRP, compared with classical carrier. 1 mL suspension was taken and 4 μL of 10 mg mL⁻¹ HRP solution was added. The samples were then oscillated at 10°C for 12 hrs. Subsequently, the samples were centrifuged at 10,000 rpm for 3 minutes, and the supernatant was collected. SDS-PAGE analysis was performed. By comparing the difference in band intensity between HRP samples without or with the addition of carrier, the adsorption amount of HRP can be determined. **b**, The SDS-PAGE image of protein supernatant without treatment (the first lane) and after loading with various materials (the 2nd to the 6th lanes).

For the immobilization of bacteriophages, alginate is the benchmark material in literature (Compr Rev Food Sci Food Saf. 2021, 20,5345-5369; Food Hydrocoll. 2021,118, 106782), and was used for comparison with our HOM-5 and HOM-15. HOM materials immobilize phages through surface absorption, while alginate immobilize phages through encapsulation, which is a different mechanism that features high loading capacity, but much slower release, thus requiring decomposition through treatment with base before use.

We compared the thermal robustness of HOM-5/15 immobilized NB1 with that of calcium alginate side-by-side, by measuring their infection efficiency following literature procedure (Poult Sci.2016, 95, 2911-2920). After treatment at 60°C for 30 minutes, the activity of free bacteriophages decreased by 90%. In the presence of

calcium alginate, the activity decreased by approximately 70%, while with the protection of HOM-5 or HOM-15, the activity of NB1 remained essentially unchanged (supplementary Fig. 53). This result shows the advantage of HOM-protected phages in thermal resistance over classical supports, like alginates.

Supplementary Fig. 53 | Lysis plaques of phage with different carrier before (A column) and after (B column) heat treatment. The weight of the three carriers is controlled to be the same.

7. The sentence on page 1, "Polystyrene microparticles, for example, are common platforms for immunoprecipitation and related techniques," along with its reference to ref. 10, seems out of place within the document. Since ref. 10 does not concern polystyrene microparticles, removing this sentence would help maintain the coherence and relevance of the discussion.

Response: We have removed this sentence along with the ref.10 as suggested.

Reviewer #3 (Remarks to the Author):

Ji and co-workers present a significant work, altering the starting molecular building blocks and conditions to produce a wide variety of microspherical architectures. I have been asked to comment particularly on the computational aspects of the work and so I focus my comments here. As most of the computational detail is contained in the SI, I have read the SI in more detail than the manuscript itself.

I think the key sentence of the authors' work is written on Line 66-68, that "... the self-assembly proceeds through threads or sheets, depending on the relative strength and directionality of pi-pi stacking and hydrogen bonding".

I agree with this statement from the authors and largely my comments center about how well they have addressed this proposal.

Response: We thank the reviewer for the thorough review on the computational aspects of our work and for the helpful comments.

My overall impression is that the authors have been both careful and thorough in the series of calculations they have done, but they haven't spent the same time on pi-pi stacking as they have hydrogen bonding.

Response: We thank the reviewer for the endorsement. We have added computational evaluation of different π - π stacking modes and included the results in Supplementary Fig. 6.

I appreciate the code provided in the SI, but I do note that it is difficult to use the code as given. It would be beneficial if the code was additionally uploaded to (e.g.) github with enough sample data to both check the code, and the results in the paper.

Response: We thank the reviewer for the suggestion. The codes we used in this study include Delaunay triangulation, dihedral angle statistics and radial distribution function. We have shared all the codes and sample data for testing on github, which are freely accessible through this link: <https://github.com/jilab2021/Hierarchical-Organic-Microsphere.git>.

My largest criticism of the computational work is that it seems to be entirely focussed on hydrogen bonding. Supplementary method 4 and the figures 4-6 cited therein, focus only on hydrogen bonds. I don't see any pi-pi stacking to compare to. I think the authors could, relatively easily, cut from their MD simulation a selection of (partly) overlapping dimers and produce something similar to SI Fig 6, but for pi-pi stacking.

Response: We thank the reviewer for this valuable comment. We extracted eight π - π stacking models from the results of MD simulations of random models and calculated their binding energies using the same method employed when considering hydrogen-

bond assembly modes. As shown in Supplementary Fig 6, we presented eight stacking models that we considered, along with their respective binding energies and Boltzmann distributions at 298K. It was observed that the stacking mode where two molecules are aligned directly exhibited the highest degree of overlap, hence possessing the lowest energy, with a distribution of 94% at 298K. This finding is consistent with the stacking mode in our constructed ideal stacking model.

Supplementary Fig. 6 | π - π stacking model of the BACT molecule. **b**, The binding energy between two molecules in different stacking models. **c**, The Boltzmann distributions of the π - π stacking models at 298 K

Smaller points:

Manuscript line 400 and in the SI, Page 3, a 2 fs timestep is cited. This seems to me quite large, normally the largest timestep ensuring energy conservation would be 1 fs.

Response: Our parameter settings were based on the literature [Langmuir 2018, 34, 6912-6921]. Here, we adjusted the calculation time step to 1fs and extended the duration of the molecular dynamics simulation to 100ns. The models before and after the simulation are shown below, and now replaced the original figure in Fig. 2j. The result shows that the hollow square tubular structure was maintained through the dynamic simulation. Shortening the timestep to 2fs does not change the conclusion of the simulation.

Fig. 2j. MD simulation for the final ideal model

SI, Page 4, middle paragraph: The enumeration of the different intermolecular hydrogen bonds is appreciated and sets out a sensible calculation strategy. I think though, at the end of this paragraph, there might be an autocorrect - I think the authors mean "*inter**molecular interactions between two molecules".

Response: We have corrected this error as suggested.

SI Fig 5: It wasn't quite clear to me whether the two dihedral angles highlighted were independent (I think so) or linked. Assuming they were independent, were their distributions equivalent?

Response: We adjusted the relevant code to statistically analyze the two sets of dihedrals, and the statistical results are presented in Fig. R3. It can be observed that both sets of dihedrals exhibit similar overall distribution patterns, primarily concentrated within 15° . As the BACT molecule is asymmetric, there exists a certain disparity between the average values of the two sets of dihedrals, with A at 12.1° and B at 13.7° . This indicates that these two angular distributions are relatively opposing, each favoring its own optimal angle tendency.

We also designated the dihedral angles A/B within the same molecule as x and y to analyze their correlation. As shown in Figure R3c, the coefficient of determination for its linear fit was only 0.0083. Therefore, we concluded that the two dihedral angles are independent. We also replaced the figure in Supplementary Fig. 4.

Supplementary Fig. 4 | b, Scheme of the dihedral angle in BACT. **c**, Distribution percentage of the corresponding dihedral angle. **d**, Distribution and linear fitting of dihedral angle A and B. The coefficient of determination (R^2) is calculated to be 0.0083, indicating the independence of the two dihedral angles.

SI Fig 6 c) and d): The dotted lines indicating the intermolecular interactions were slightly difficult to see - highlighting them in a brighter color would help a lot.

Response: We have revised the Supplementary Fig. 5c and d by changing the original blue dashed lines to thicker orange lines as shown below:

Typographical errors:

Manuscript Line 74/75: "With respect to function, we applied these sixteen HOMs displayed..." I think two sentences got mixed together here. If the words "we applied" were deleted, I think it would say what the authors mean.

Response: We thank the reviewer for the correction. We have deleted "we applied" as suggested.

SI Page 5: "spaces group" -> space group

"force flied" -> force field

Response: We have fixed these typos as suggested. Thank you again for all the great suggestions.

REVIEWERS' COMMENTS

Reviewer #1 (Remarks to the Author):

The manuscript has been much improved. As shown in CO₂ sorption isotherms, it is not suitable to refer the present materials as porous materials. However, systematic construction of spherical objects based on organic molecules presented in the manuscript is very important and is well-investigated. The reviewer feels that the manuscript is now acceptable for Nature Communications.

Reviewer #2 (Remarks to the Author):

The authors clearly addressed the comments and such revised manuscript may be accepted for publication.

Reviewer #3 (Remarks to the Author):

My concerns have been addressed in the revised manuscript. I thank the authors for their work.

Reviewer #3 (Remarks on code availability):

I viewed but did not install or run the code as I don't have access to matlab currently. The code looks sensible and is in a format that can be 'dropped in' by anyone with the right license.